# Witnessing Self-Affirming Moments in Persons with Dementia While Interacting with Therapy Dogs: A Case Report

**DOI:** 10.3390/ani14243620

**Published:** 2024-12-15

**Authors:** Carolyn Brooks, Colleen A. Dell, Darlene Chalmers, Ben D. B. Carey

**Affiliations:** 1Department of Sociology, University of Saskatchewan, Saskatoon, SK S4N 5A5, Canada; carolyn.brooks@usask.ca (C.B.); ben.carey@usask.ca (B.D.B.C.); 2School of Public Health, University of Saskatchewan, Saskatoon, SK S7N 5A5, Canada; 3Faculty of Social Work, University of Regina, Regina, SK S4S 0A2, Canada; darlene.chalmers@uregina.ca; 4St. John Ambulance, Regina, SK S4T 0C8, Canada

**Keywords:** animal- and canine-assisted support programs, animal-assisted support program, therapy dog handler, caregiver, physical movement, psychological shift, time transition, connection, intervention

## Abstract

Alzheimer’s and other dementias affect people worldwide, causing anxiety and stress for both diagnosed individuals and their caregivers. We studied the cases of four people with dementia who interacted with therapy dogs in different settings: at home, in a hospital, and in a psychiatric centre. Based on the insights of two experienced therapy dog handlers involved in these cases, we found that therapy dogs help in four main ways: encouraging movement, changing feelings, altering the sense of time, and fostering connections. Therapy dogs seem to help create positive interactions, turning difficult times into moments where people appear to feel good about themselves. Therapy dogs show promise as a helpful intervention for people with dementia and their caregivers, but more research is needed.

## 1. Introduction

Across the globe, individuals, families, and communities are impacted by Alzheimer’s disease and other dementias, and the number of people diagnosed is projected to increase with worldwide aging population trends [1]. At present, an estimated three-quarters of a million people in Canada are diagnosed with dementia [2]. Family members and workers in the health care system frequently observe individuals living with dementia experience unpredictable and difficult moments involving anxiety, stress, fear, and delusions [3]. A recent Canadian study of caregivers of seniors found 26 percent experienced symptoms of distress, and this increased to 45 percent for individuals caring for seniors living with dementia [2]. The dementia literature frequently refers to the illness as causing a change in the affected individual’s sense of self [4]. However, just as there are difficult moments associated with dementia for those impacted, so too are there hopeful moments with common experiences of love, happiness, and joy [5,6]. The literature recognizes that individuals with dementia take pride in themselves in such moments [4]. It is these experiences that require more consideration.

This case report explores how individuals who care for people with dementia can help redirect difficult personal and interpersonal moments towards hopeful moments in the presence of therapy dogs. In Canada, therapy dog interventions are commonly offered to individuals challenged with dementia. Although evidence of their beneficial impact is growing, almost no research or literature considers their potential for self-affirmation. Therapy dog visiting appears to facilitate positive social interactions that can help redirect difficult moments to moments defined by self-affirmation or experiencing a positive sense of self. This paper presents the cases of four individuals diagnosed with dementia and interacting with a therapy dog in one of three environments; the cases are presented from the perspectives of two attending therapy dog handlers. Self-affirmation theory surfaced as a guiding framework for explaining their observations and experiences. The identified themes offer some potential insight into the poorly understood process of self-affirmation for individuals experiencing dementia when interacting with a therapy dog. Therapy dogs are presented in this case report as a promising psychosocial intervention for individuals with dementia and their caregivers. We conclude by considering the implications of this for future research and practice attention.

We note that animal-assisted interventions, also referred to as animal-assisted human services, are comprised of animal-assisted treatment, animal-assisted education, and animal-assisted support programs, the latter being the focus here. Animal-assisted support programs are “aimed at supporting and enhancing the well-being of humans” [7]. One example is animal visitation services (formerly known as animal-assisted activities), with dogs being the most common animal involved. This typically entails community volunteers visiting with individuals who do not have access to a dog. Therapy dogs, also commonly known as comfort or wellness dogs, are friendly family pets who enjoy visiting with people in the community alongside their human handlers. Both the dog and human handler are tested for their suitability in this role by community organizations. The most prominent therapy dog organization in Canada is St. John Ambulance, from which volunteer teams visit a variety of settings to offer comfort and support; the leading environment for visiting is senior care. It is important to note that handler, therapy dog, and therapy dog-participant welfare are key components of the testing and program [8]. We use the terms ‘therapy dog, ‘therapy dog program’, and ‘therapy dog team’ herein, as they are preferred by the St. John Ambulance Therapy Dog Program.

## 2. Background and Literature Review

### 2.1. Self-Affirmation Theory

Self-affirmation theory is well-established in the psychology field, first identified by Steele [9] as important to protecting one’s self-image, worth, and integrity. It has two main premises: having a positive sense of self is important to a person’s overall health and well-being, and people aim to restore their self-integrity if it is compromised [10]. According to the theory, people are motivated to maintain self-regard and see themselves as capable, having free choice, competent, controlling their own life choices, stable, and good [11]. The theory focuses on motivations towards self-affirmations, factors that reduce threats to the sense of self, and supports and reduces barriers to effective self-affirmation [10].

Self-affirmation interventions are noted in some non-pharmacological literature as relevant to caring for persons living with dementia. Non-pharmacological interventions range from providing psychoeducational interventions for caregivers [12] to meaningful activities based on the interests of the person living with dementia and their support persons. Many of these include a focus on self-affirmation. For example, Baird and Thompson [13] suggest the disrupted sense of self experienced by people living with dementia can be alleviated with music. Swall et al. [14] (p. 1) provide an overview of person-centered methods including “validation therapy (Söderlund et al., 2012) and person-centred conversations (Hedman et al., 2016)”, demonstrating how these therapies may bring back a sense of self. Person-centered methods can trigger memories or “so-called reminiscence (Nordgren & Asp, 2019; Woods et al., 2005) and temporary presence of mind in the moment (Normann et al., 1998)”, increasing perceptions of self-remembered [14] (p. 1). Other person-centered approaches that focus on bringing back a sense of self integrate robots designed for wellness. Moharana et al. [5] (p. 461) discuss the utilization of “Robots for Joy” to help individuals living with dementia focus on moments that bring them happiness. Other examples of meaningful non-pharmaceutical approaches to engage a sense of self include exercise, therapeutic touch, music therapy, light therapy [12], and therapeutic gardening [15].

Non-pharmacological approaches, as alternatives to, or in conjunction with, pharmaceutical responses, show promise for supporting people living with dementia and their caregivers, but some debate remains. Some authors cite a lack of supporting evidence, often due to difficulties regarding methods [12]. While self-affirmation theory is valued for its richness in scope, simplicity, and testable theorems [16], limited research has sought to understand how animal-assisted support programs, and canine visiting specifically, may be linked to self-affirmation for people living with dementia.

### 2.2. Promise of Canine-Assisted Support Programs as an Intervention

Research on animal- and canine-assisted interventions among people living with dementia is showing promising results [17,18,19], but is still in its infancy [20,21]. The integration of therapy dogs shows diverse benefits related to quality of life [20], increased physical and social activities [18], and decreased turmoil and unrest [18,19]. Canine-assisted support programs have demonstrated social [14,22,23], spiritual [14,24], psychological [25,26], cognitive [27], and physical [18,28,29] health and well-being benefits for people living with dementia. For example, in their review of 22 quasi-experimental and randomized controlled trials, Posadas et al. [18] (p. 102) found “a significant increase in a variety of positive social behaviours such as smiles, looks, tactile gestures and verbalizations, i.e., DAT [dog-assisted interventions] enhances pro-social behaviour.” Physical benefits include a slowing of dementia symptoms [18], a reduction in stress, better pain management, lower blood pressure [14], cognitive behavioural improvements, memory recall, and healthier choices, including nutritional choices [30,31,32].

While some research remains hesitant to confirm positive impacts, this is buttressed by suggesting negligible adverse effects [18,29], with the slight concern raised by Swall et al. [14] that interactions with therapy dogs may increase sadness. Agreed throughout the literature is that more studies are needed [18,20,33,34,35,36], including mixed-method studies [37] and work specific to potential side effects [18]. More well-designed and randomized control trials and quasi-experimental studies are also called for to record more consistent results [18,20]. Qualitative studies could also help to answer questions about how beneficial impacts are arrived at, or not.

Qualitative research specifically adds additional depth to understanding the nature of the impact of animal-assisted support programs among persons living with dementia [18,24,34,35], but is sparse. Findings to date provide in-depth accounts of human–animal interactions and how these may lead to feelings of closeness, joy, being in the moment, an awareness of the past and present, and intimacy with an animal. For example, Olsen et al. [38] used video recordings of behaviours during animal-assisted interventions, finding them to be engaging and possibly health-promoting. Using a phenomenological hermeneutic method to determine benefits to care, focusing on lived experiences, Swall et al. [14] (p. 4) videotaped interactions between persons with dementia and therapy dogs at multiple nursing homes in Sweden and reported increased self-worth, as well as joy, empathy, and altruism: “The presence of the dog brings joy and closeness for that moment, and also results in a sense of appreciation and moment of happiness in daily life”. Dell et al. [39] and Swall et al. [40] (p. 89) further reflect on how therapy dogs connect the person with memories and their “relationship with other dogs in life”.

Other qualitative studies feature interviews with health care professionals and dog handlers. Gundersen and Johannessen [15] (p. 109) conducted in-depth interviews with nurses and dog handlers in Sweden and found positive impacts for both: “The visits made the dog handlers feel useful and gave them a sense of joy and meaningfulness… the nurses… were impressed by and thankful to the dog handlers who gave their time as volunteers.” This can intersect with the well-established view that system caregivers who feel valued by individuals and the environments in which they work generally offer more comprehensive and compassionate health care [41,42]. Swall et al. [24] interviewed dog handlers to understand the impact of therapy dogs on people living with dementia nearing end of life. The results indicated very positive impacts, including an intimacy and trust in the dog to share existential thoughts related to their own passing, as well as the provision of comfort.

## 3. Methods and Materials

### 3.1. Reflexivity Statement

In response to the need for an in-depth understanding of the possible benefits of canine-assisted support programs among people living with dementia, this case report draws on the reflexive experiences of two experienced therapy dog university researchers who have over 25 years of combined therapy dog visiting experience in Canada, including among countless individuals diagnosed and living with dementia. Dell has been a handler with the St. John Ambulance Therapy Dog program since 2013, and Chalmers since 2014. Their therapy dog practice and research has served a wide variety of populations, including individuals diagnosed with dementia, and within diverse settings, such as prisons, psychiatric facilities, hospitals, hospice, veterans’ long-term care residences, and in emergency community responses. They are also the co-directors, alongside Dr. Linzi Williamson, of the PAWSitive Connections Lab at the University of Saskatchewan; their research lab focuses on exploring interactions between animals and people, with specific attention to the intersecting health and welfare of both. They are also trainers of two service dogs each for a community-based project examining the beneficial role of service dogs in the lives of veterans diagnosed with a traumatic brain injury [39]. Their work has been recognized with medals and awards of honour, including appointment to the Most Venerable Order of the Hospital of St. John of Jerusalem, as a part of Canada’s honours system.

The first author, Carolyn Brooks, is a critical sociologist interested in social change, and with an extensive background in community-based research methods. Brooks applied her expertise with the interpretivist method, focusing on the subjective experiences of people, to conduct the focused conversation for this paper. She does not have a background in either dementia studies or animal-assisted support programs. The last author, Ben Carey, has been a project manager and researcher in the PAWSitive Connections Lab since 2018, and is currently the coordinator of the St. John Ambulance Therapy Dog program in Saskatchewan. Through these roles, Carey offers in-depth and practical insight into the canine-assisted support program field.

Each author brings potential reflexivity bias to this case report, including the fact that Chalmers, Carey and Dell are invested in the field as volunteers or employees. Brooks is not. As a novice to the field, Brooks undertook the focused conversation in an analytical, inquisitive, and critical manner. This is evident, for example, in the fact that Dell and Chalmers had not addressed self-affirmation theory in any of their prior work. Including Carey in the validation process helped to minimize the effects of potential biases. Given the critical social research backgrounds of all authors, each has also been educated to critically question and assess the social environment surrounding them.

### 3.2. Therapy Dog Interactions

Three environments, encompassing Chalmers and Dell’s experiences with therapy dogs with four individuals living with dementia, are the focus of this case report. The first, in a family home, involves a woman (Maggie, pseudonym) in her 80s diagnosed with Alzheimer’s disease. Chalmers and her four-year-old, energetic Labrador Retriever “Ruby” were regular visitors over a two-year period. Visits averaged one hour in length. Second, Dell visited weekly with a dementia patient for an hour with “Subie”, a four-year-old low-key Boxer breed, in a secure unit of a hospital for patients at risk of fleeing. Over a four-month period, Charlene (pseudonym), a middle-aged woman, was waiting for a placement in a long-term care residence and was periodically confined to a restraint chair in her hospital room. Third, both Dell and Chalmers visited Tim (pseudonym) with therapy dogs Ruby and Subie, while he waited in a secure hospital dementia unit for placement in a longer-term care facility. Tim was in his 80s and was visited weekly by one of the two therapy dogs, for between 30 and 60 min, for approximately four months. And last, John (pseudonym) was a long-term, middle-aged patient at a psychiatric facility and displayed increasing dementia symptoms over an 8-month period of weekly 30 min visits from therapy dogs Subie, Ruby, and “Kisbey”, a 6-year-old energetic Boxer breed. Further identifying characteristics for Maggie, Charlene, Tim, and John are not shared here, to maintain their anonymity.

### 3.3. Data Collection

This case report is the result of a focused conversation [43] between first author Brooks and co-authors Dell and Chalmers, which explored their perceptions of their own research and experiences with and observing therapy dogs interacting with people living with dementia. Following Spee’s [44] method of collecting information through focused conversations with groups who share a common experience, we aimed to uncover what Dell and Chalmers observed and their reflections and interpretations of their experiences.

The focused conversation method progresses from observing concrete experiences to reflecting on them, generalizing, and forming more abstract concepts [43,44]. This cycle repeats, guided by questions aligned with the ORID technique. This technique parallels the goal of a case report, which in this case is to expand the dementia and therapy dog knowledge base. The method allowed four phases of data collection:Objective Phase (O): This phase centers on questions and discussions about the four case studies, focusing on what participants believe these studies reveal or fail to reveal about the interactions between therapy dogs and people living with dementia.Reflective Phase (R): Participants reflect on their own and each other’s responses, adding personal and cultural significance to the cases.Interpretive Phase (I): This phase allows participants to draw conclusions and interpretations from the cases.Decisional Phase (D): Discussions in this phase revolve around how participants envision advancing in research and action [43,44].

This qualitative conversation was recorded using Zoom, a videoconferencing platform with enhanced recording capabilities. This was transcribed verbatim by Brooks, to ensure accuracy and emotional context. Because this method included studying researchers’ secondary reflections of their practice, ethics approval was not required. Chalmers and Dell provided recorded verbal consent.

The four cases reflected upon in this case report emerged naturally within the conversation as points of focus, and were not pre-determined. All four cases were discussed amongst Dell and Chalmers in past conversations, as they did with numerous other individuals they visited with the therapy dogs, and in a sense this “kept these cases current”. The attention given to them also reflects the impact these cases had on Dell and Chalmers. The earliest date of a therapy dog case visit was 2016 and the latest 2021, just months before the conversation took place. The other two case visits occurred in 2017 and 2018.

It is important to point out that this case report follows a traditional format reflective of the health sciences. This means that as a research design, a case report “describes important scientific observations that are encountered in a… setting to expand our knowledge base” [45] (p. 126). Case reports are not as in-depth and formulated as case studies, which are common to the social sciences. Case studies are elaborate, multi-faceted approaches to examining complex issues in great detail and in real-life settings. Case reports, on the other hand, are descriptive, do not answer a research question, deliver a clear message, and offer insightful information. Case reports are known for initiating new research studies based on their unique observation, but these are not “confirmed unless we conduct further confirmatory quantitative experimental or observational study designs” [45] (p. 127). Additional methods would apply with a social science-focused study. Within the medical community, where they are prevalent, case reports are identified as the “tail of the hierarchy of evidence-based medicine” [45] (p. 127), but yet have important value.

### 3.4. Data Analysis

Dell and Chalmers’ words are conferred as quotes here to present what was witnessed in their roles as therapy dog handlers and as researchers. They offer initial reflections on how their two-fold positionality allows them to “witness” as opposed to simply “observe” moments between the therapy dogs and the people they visit. Dell and Chalmers each recognize their unique dual position, allowing moments with the therapy dogs to integrate with their research knowledge of animal-assisted support programs. This makes them informed witnesses to interactive moments between the therapy dogs and individuals with dementia with whom they are visiting. This is illustrated in Dell’s observation:
*I like [the idea of] witnessing, because… as a researcher we can then take what we observe and experience and integrate it with where we are at in scientific understanding, and move the field forward, because we really are witness to these intimate experiences and these heartfelt experiences in these real, genuine, honest, vulnerable moments. We get to witness the dog doing exactly what the dog does.**(Dell)*

Other forms of work, such as ethnographies, align with this approach. The work of James Baldwin, for example, discussed by Ince (2023), notes that one’s role as the researcher in a social system “shapes how one knows and does not know, what one sees and does not see, and how one imagines freedom and justice”. This is particularly relevant to this case report.

The focused conversation between Brooks and Dell and Chalmers was nearly two hours (110 min) in length, with topics covering their observations and experiences, and interpretations, based on their academic research, of the impact of therapy dogs, generally, and of individuals experiencing dementia, specifically. Brooks conducted data analysis and interpretation according to Fritzen-Pedicini et al. [43] and thematic analysis according to Naeem et al. [46], with the arising themes confirmed by Chalmers and Dell, as well as Carey, in follow-up discussions and the writing of this case report. Our data interpretation approach was grounded in the ORID technique (discussed above) and a five-step thematic analysis [46]. First, during transcription, we delved deeper into the data to identify emerging themes. Second, we selected keywords and short phrases (codes) to encapsulate core messages and their significance. Third, we organized these themes into meaningful relationships, linking them to the research topic on therapy dogs and individuals living with dementia. Fourth, we conceptualized how these themes fit together and reflected on their alignment with the existing literature and theories on therapy dogs, dementia, and self-affirmation theory. The fifth step involved understanding the significance of these findings and their contribution to the field.

## 4. Results and Discussion

The observations and experiences, or more in-depth witnessing according to Chalmers and Dell, suggest interacting with a therapy dog in a canine-assisted support program may assist in providing experiences for those living with dementia that allow for increased self-affirmation. This is experienced through (a) physical movement, (b) a psychological shift, (c) a transition in the experience of time, and (d) connective interactions.

### 4.1. Self-Affirmation and Physical Movement

Therapy dog visits were often in spaces that could potentially be viewed by the persons with dementia as being less restrictive and controlled. Visits with the therapy dog commonly meant they were able to leave restricted areas in the hospital, for example, and go for a walk and sometimes sit outside. This form of physical freedom and expression of personal agency is oftentimes limited when people are institutionalized or somehow restricted because of their illness. Dell describes the power she witnessed in this physical movement for Charlene and Tim. In both cases, their environments were locked for safety, and the therapy dog visit allowed them the freedom to move. Dell describes the difference for Tim between being in the hospital ward, which was often quite loud, with residents at times crying and yelling, and a quiet walk with the therapy dog, Subie, in the hallway off the unit: “his agitation visibly reduced and he increasingly expressed joy”. For Charlene, visiting with therapy dog Subie allowed her to go off the unit on which she was sometimes restrained in her room in a chair. Off the unit, “you could sense the tension ease from her body”, according to Dell. “I never had a concern that she would run away when she was walking with Subie”.

Whereas positive self-affirmation can often lead to more physical activity and socialization within a general population [47], here we suggest physical activity and the ability of some forms of movement can potentially enhance self-affirmation amongst those living with dementia. Self-affirmation theorists point to the importance of physical activity (Jessop et al., 2014, cited in [47]) and its relationship to autonomy. The opportunity to move freely and unrestrained is important for a sense of dignity. Persons with dementia and other individuals with cognitive impairment may display some aggressive behaviours that result in restraint or lockdown; yet disruptive behaviours can worsen with the use of physical restraints and when people with dementia are left in isolation [48]. Non-pharmacological interventions, such as interactions with therapy dogs, may provide even a brief opportunity for less restraint, with the increase in physical movement potentially being viewed as self-affirming.

Self-affirmation through physical movement is also noted in other studies focused on the inclusion of therapy dogs. Swall et al. [24] found therapy dog responsiveness gave the person experiencing dementia motivation and energy to be more physical, using more of their body. Notably, they suggest that the release of endorphins and other hormones (e.g., oxytocin, prolactin, dopamine) that occurs when the person with dementia pets an animal can lead to improvements in cognitive functioning, decreased feelings of anxiety and tension, and increased feelings of joy.

### 4.2. Self-Affirmation and a Psychological Shift

This second theme relates to a perceived psychological shift to having power and controlling outcomes. This was witnessed through the individual living with dementia and the therapy dog handler sharing a leash to guide the dog. This created opportunities for the therapy dog to focus their attention on the person and demonstrated the person’s likeability and autonomy. Dell describes the empowering interaction and the impact of having the therapy dog’s attention, for both Tim and Charlene:
*The individual has the dog [Subie] on the leash, and is walking with the dog side by side, and if the dog goes too far ahead, they’ll say, hey, calling him [Subie] back, and the dog listens. Imagine the autonomy we have in having the dog listen and respond to you… There is fulfilment for individuals when the dog listens to them.*

Dell and Chalmers both said the therapy dogs brought out parts of people’s personalities they may not have otherwise seen without the therapy dog’s presence, which they view as person-affirming. Chalmers spoke about Maggie, who showed a “softer side of herself” during interactions with Ruby:
*She’d be talking with the therapy dog, and she called herself grandma—‘so do you want gramma to… do you like gramma…you’re the bravest’… [Maggie] would say things like, ‘I think she really likes me, yes, she does’, and ‘she’s following me into the kitchen. Yes, she is. So, she likes me a lot’. And I [Chalmers] was able to say, ‘Yes, she does’. So that’s affirming.*

Dell added that therapy dog handlers do a lot of person-affirming as well: “I think we do that all the time.” She describes how she often responds when others are interacting with her therapy dog: “No matter what the dog is doing, [I can say], ‘oh, wow, she’s really looking at you,’ and they exaggerate a little bit more…. That is person-affirming.” In this way, both Dell and Chalmers helped to explain how the interactions with the dogs, not just the words, showed people they were cared about and reminded people about their own competencies, kindness, and skills. While this was perhaps reinforced by reminders from the handlers, e.g., “wow, she’s really looking at you,” the actions of the dogs following them, looking at them, coming when called, etc., were the actual facilitators.

Expressions of kindness, care, and even love also showed up in more obvious ways, such as direct expressions of intimacy, including saying “I love you” to the therapy dog or sharing a secret. Dell adds this level of intimacy takes longer, but is common, and she witnessed it between John, who experienced dementia during his incarceration at a psychiatric facility, and therapy dog Kisbey:
*You know, being able to say, ‘I love you,’ like in a socially acceptable way. And that doesn’t happen right away. That takes a long time. And then telling her [Kisbey] something that I probably wasn’t supposed to hear… it’s probably something they needed to share, as they were not going to be around much longer in the program.*

Chalmers reflected on the importance of intimacy and closeness and that the therapy dog is another living, sentient, thinking being that seems to allow for that level of closeness. She too visited with John, alongside therapy dog Ruby:
*It is different than other kinds of support activities… because it’s another living individual that thinks and feels, and that the individual living with dementia can be with in ways that they cannot be with, maybe even family… where they can be close, right? Especially if institutionalized. There’s intimacy that can occur and is socially accepted in public.*

Studies with persons living with dementia demonstrate how the self is affected and the importance of experiencing oneself as valuable [4]. Interaction with the therapy dogs seems to create experiences for recalling one’s competencies and skills, as well as how much individuals are liked, appreciated, cared for, and even unconditionally loved.

Reflections by Dell and Chalmers support studies that suggest interactions with therapy dogs can bolster psychological well-being and be self-affirming. Psychological impacts of animal interactions on people’s inner lives are described by Swall et al. [14], who contend that lucidity is experienced by being more present with the dog. They suggest caring about the dog’s best interests makes people with dementia more expressive of who they are:
*It appears that the person emerges from the shell created by dementia and may show a fragment of his or her person because of being empowered in the dog’s presence. Together with the dog, they possibly show their innermost self. They liven up and demonstrate the abilities and resources of a whole human being and know that they can take care of the dog using their current faculties.*[14] (p. 6)

In a similar way, Chalmers and Dell witnessed how leash walking led to a sense of self-control and being, drawing on personal skills and power.

The literature points to interactions with the therapy dogs as increasing happiness and positive sense of self [14]. Dogs follow, stare, and snuggle, creating moments where persons with dementia can be reminded of being loved. The dog’s presence allows the dog handler to be directly involved in self-affirmation as well; this aspect has not, to our knowledge, been noted elsewhere. Dell and Chalmers indicate this could be initiated and/or facilitated in the interaction with the therapy dog, alluding to the possibility that these interactions could be more intentionally designed to be self-affirming.

### 4.3. Self-Affirmation and a Transition in the Experience of Time

Interacting with a therapy dog also meant possibly being temporarily removed from some of the symptoms of dementia, both because the focus is on the ‘here and now’ with the dogs and because this shifts one to happier or different times in animal-related memories, including with pets and farm animals. Chalmers talked about this as a movement of “time, with a focus on the present time.” This type of transition in time could possibly allow for a decrease in some common dementia symptoms, such as anxiety of the unknown, not remembering, fear of having people steal from you, and the terror associated with not recognizing where you are. For example, Chalmers talked about how Tim would become elevated, worrying about people taking his belongings. She said therapy dog Ruby helped bring him out of moments of fear and delusion, re-directing him back to the here and now, and thus alleviating fear and other symptoms:
*These moments transcend space and time because they’re connected. Space can be where the individuals are at… and bringing it back to the present moment. And so the time piece for me is being able to do that, it’s being very present, even in one’s thinking in their body in space. But just being in that moment; that is freedom*.

This transition in time—being present—was also discussed through the idea of memories, brought on through interactions with the therapy dogs. Dell spoke about interacting with Charlene and noticing that memories and reminiscence would shift experiences to happier places:
*We talked about the dogs when they were a kid, and these relationships with the animals, and I… learned all about their mom and dad and siblings and select experiences growing up… I could see a little bit of de-escalation, a little bit of memory to those happier times and places in their lives.*

Chalmers defined this type of movement in time as a type of existential experience and affirmation of one’s existence:
*… reminiscing to me is potentially an existential experience for people because they’re able to return in memory or be in moments in time… many of the stories that we’ve heard from people are really great times… they’re thinking… back to times of having a dog at different points during their life.*

Dell and Chalmers discussed the increased sense of self seemingly achieved by remembering other times in one’s life and/or by the relief in being in the present, focusing on current experiences with the therapy dog. This may increase a feeling of being comfortable in one’s own body and a confirmation of self. Their reflections recognized the benefit of the therapy dogs in terms of directing individuals away from perceived difficult feelings, such as fear or a loss of self, possibly by bringing up memories of times before dementia. This impact of the therapy dogs would create opportunities for increased self-affirmation.

To self-affirm, research suggests people will orient towards qualities and/or memories of self that align with the larger views of themselves. Harris et al. [49] (pp. 590–591) explain that self-affirmation is partly done by “reminding the individual that his or her (their) self-worth is not solely dependent on the threatened domain.” As such, “self-affirmation can reduce the extent to which information in that domain threatens the self and enable the individual to engage in more open-minded appraisal of potentially threatening self-relevant information.” They suggest a person may, e.g., “remind herself that she is a good mother or daughter or a talented musician.” For someone living with dementia, opportunities for memories are viewed as self-affirming [14,49].

Swall et al. [14] (p. 6) take this further, showing how moments with an animal may not only be linked to bringing up memories, but to shifting definitions of self, from one experiencing dementia to the self before the decline in cognitive abilities: “These memories may also, albeit temporarily, result in the person with AD [Alzheimer’s disease] behaving like a confident, healthy person.” Dell and Chalmers both spoke about the ability of the interactions with the dogs to bring forward stories and memories related to self, including reminders of pets the persons with dementia or their families had lived with. This, they put forward, appeared to be self-affirming, with respect to how individuals saw themselves in that moment, and potentially how others saw them too. They added that these memories may shift the way a therapy dog handler and possibly others (staff, nurses, etc.) view the person with dementia, therefore enhancing or changing relationships. The person is recognized and defined by other moments in time, allowing further understandings of each other’s lives and a possible depth to a relationship that was previously primarily defined by the person since the onset of dementia. This would be self-affirming, given the need as human beings to know who we are but to also feel recognized by others for who we are [4].

While much research shows the impact of canine-assisted support programs on pro-social behaviours and interactions (e.g., [50]), Chalmers and Dell add further insight: the handler is not able to think of personal things very much or to be hurried. Their reflections suggest the dog makes everyone more present and that interactions may become less encumbered as a result. While the dog and handler are guiding the moments away from fear or other symptoms of dementia, the dog in this instance, by the honesty of their presence, is shifting the moment of the handler, as well as the person with dementia. Considerable information and advice are available about how to be less distracted within caregiving roles with people living with dementia, including how to actively listen, not worry, and focus on the individual (e.g., [51]). The insights here suggest the benefits of therapy dogs for everyone sharing the space in terms of being present.

### 4.4. Self-Affirmation and Connective Interactions

One’s sense of self is affected not only by pride in one’s abilities and self-affirmation, but also in the ways others react to us [4]. Dell and Chalmers describe an increase in connective interactions of people living with dementia, caregivers, dog handlers, and others. The therapy dog became a buffer for engaged and energetic communication, and interactions were viewed as more unencumbered for individuals living with dementia and for those with whom they were interacting.

The therapy dog’s presence facilitated meaningful interactions with others, not just because of the time spent with the dog and handler, but because the dog’s presence offered a bridge or buffer to communication with others. Dell spoke of the therapy dog making communication more spontaneous and meaningful:
*They’re [the therapy dog] a bridge, right? They’re the connection. It takes pressure off me in some ways. Not that I feel a lot of pressure, but what do you talk about? And then for that person, even more pressure if felt, possibly. They’re not put on the spot [when the therapy dog is there].*

Chalmers spoke about the dog being a ‘social lubricant’ in any number of interactions, including in time spent with Charlene, Tim, Maggie, and John:
*They open all the doors, windows to whatever will come and to whatever is going to happen in that interaction. The therapy dog’s presence allowed for connections even if no actual words were spoken.*

Dell and Chalmers suggest the dog takes pressure away from interactions simply by being the focus of most everyone sharing the space. Chalmers, for example, talked about what this looked like in the hospital setting where Tim was residing:
*I think about walking with [Tim] out to this little main sitting area typically where we come in an entrance with a number of people sitting and watching the comings and goings. They might not be saying anything, but they are part of the interaction.*

Dell adds that, when the therapy dog is present, most everyone seems to be focused on the dog, and it becomes a connection for the people who are with the dog. It is also a bit of a distraction: “It’s like a connection for them with the other people in that kind of way… It’s also a little bit of entertainment… [otherwise] they were all glued to the TV because what else do you do?”

Dell and Chalmers’ reflections about connective interactions were deepened by witnessing what they both identified as unencumbered interactions. In part, the dogs allowed people involved in the interaction to be more fully present and less distracted, simply due to the presence of the therapy dog in the room. Dell explained that support workers, family, and health care staff seem to have very good intentions, but they are not always fully present. This is not a critique, but rather an acknowledgement that we all may be distracted by our own needs. Dell makes this connection first, explaining how people come with their own time frames, expectations, and needs. Dogs, she reflects, have none of this, and it becomes more about the moment and the person:
*You have all the workers in these places who are nice people, but they’re probably not a hundred percent present. They probably get overwhelmed. I do too sometimes… I bring all that I am experiencing that day with me, whereas the dog doesn’t bring a lot, no expectations. He’s not going to interact with you with a goal in mind for your own well-being, the dog’s goal is more like wanting to be pet.*

Chalmers agreed, and added that her therapy dog forces her to be more in the moment. She gave an example related to how long it could take for Maggie to get up and get ready for the dog walk. While this may have led to a focus on, or even frustration about, the time it was taking for her to get ready, Chalmers had to slow down and pay attention to Ruby:
*I was paying attention just to where my dog was at the moment, too, and being present with her. And so, everything sort of grinds back… just to be here in these moments, whatever they are, however long it takes to get where we’re going… I can’t be thinking about what’s going on at home, or my grocery list, or about what happened at work. I must be here in this moment.*

Handlers need to be focused, as they have the added responsibility of making sure the therapy dog is safe and healthy and of guiding a relationship between the dog and the person they are visiting. This kind of intentional focus does not allow attention to sway; this means more focus on the present moment for both the handler and the person with dementia they are visiting.

Dell and Chalmers also witnessed a shift in perspective, based on the presence of the therapy dog and impact of the handler facilitating moments with the dogs, which was described as powerful. Chalmers talked about how the interactions with the therapy dog even created opportunities for moments of awe and heightened appreciation for the dogs that seemed to be shared by everyone involved. Dell said you could see the change in people’s faces when they came to say hi to the dog. Chalmers and Dell said staff and others in the spaces may view people with dementia in a different way when they are with the dogs. This was experienced more existentially, for example, by shared eye contact between the dog and those they visit; Chalmers recognized it as an understanding of an inherent beauty and value of the therapy dog interaction:
*This is beyond human. It’s powerful. It is connection, and I think, wow! You’re [the animal] so amazing… the way you are in the world, and the way… you make sense in the world… You see the person that you’re with and they see that too. You’re having that moment, the wonder of that beautiful animal that is in front of us and it’s the wonderment. Maybe that’s the experience of those individuals as well… its awe-inspiring.*

Therapy dogs are often viewed as being there for an individual’s well-being. In this case, connective interactions with the therapy dogs themselves can be beneficial, and interactions with others via the therapy dogs can become a buffer for communication. The shared benefit from the dog’s presence and from developing relationships beyond the dog is central to experiencing self-affirmation.

## 5. Suggested Next Steps

Increasing positive, connective social interaction is an important theme in much of the research on therapy dog interactions with persons with dementia (see, e.g., [13,18,19,20]. For example, Postadas et al. [18] (p.101) review 22 studies on therapy dog interventions in nursing homes, and find increases in social interactions as well as “laughing, smiling, touching and talking,” which they attribute to “being encouraged to visit and interact/touch the dogs.” Travers et al. [29] (p. 214) suggest canine-assisted interventions are successful, specifically because they address unmet needs of persons with dementia that often underlie symptoms related to behaviour and well-being, such as social interaction: “… dog-assisted therapies… [introduce] activities that provide meaningful activity, stimulation, pleasurable social interaction, and comfort through physical contact.” They further note that therapy dog interventions can help decrease agitation and other symptoms and increase pro-social behaviours and calmness with others. The literature on human–animal bonds generally speaks of the benefits for human health [52].

Having a positive sense of self is linked to experiencing ourselves as being valuable and having a sense of who we are, and who we are with other people, yet those living with dementia “have been regarded [as] losing their self” [4] (p. 205). The loss of memories is only a part of one’s life experiences, and reflections in this case report support the idea that our own agency or self-affirmation is not entirely compromised with memory impairment [53]. Connective interactions with therapy dogs have the potential to enable persons with dementia to re-engage and achieve a degree of self-affirmation, in turn reminding the person of their larger self.

In this case report, we do not attempt to define what self may mean for individuals diagnosed with dementia; we do, however, suggest the experience of dementia does not mean a complete loss of self. Rather, we consider how dementia may disrupt a person’s sense of self as well as considering the importance of self-affirmation in bolstering feelings of adequacy and personhood. Harris et al. [49] explain that “to self-affirm, people turn to the qualities that are central to how they see themselves. Self-affirmation involves bringing to mind any self-conception or image that bolsters or restores the individual’s sense of being ‘competent, good, coherent…’” (Steele, 1988, p. 262; cited in [49] p. 590). This is of importance for people living with dementia.

We recognize the diversity of experiences among individuals living with dementia, and the literature’s support of non-pharmaceutical approaches for self-affirmation. Our case report contributes to this current understanding, adding insight into that process of interacting with therapy dogs to not only disrupt difficult moments, but also facilitate understanding about how caregivers of people living with dementia may redirect difficult moments to more positive moments of experiencing such feelings as joy, pride, and care in the presence of therapy dogs. Therapy dog visiting appears to facilitate positive social interactions that help redirect difficult moments to moments defined by self-affirmation or experiencing a positive sense of self. The literature related to dementia and therapy dogs supports our four themes individually and their bridging together for a deeper understanding of the important role of self-affirmation, and the possible facilitating role of therapy dog visits in experiencing it among people living with dementia.

Based on the witnessing of observations and experiences by our two therapy dog handlers/researchers, and the existing therapy dog, dementia, and self-affirmation literature, we suggest that future studies with therapy dog handlers visiting individuals living with dementia could foremost explore the following: if/how handlers help to direct moments of time (change) away from fear and other symptoms of dementia to being in the present moment with the dog/affirming self; if/how the therapy dog affirms the person by perceiving to be liking/loving them unconditionally; and if/how the handler can encourage this and use it as a further opportunity to remind the person how liked and loved they are, supporting their ability to self-affirm. We also suggest that studies be designed with methods that allow for an in-depth understanding, for example, of experiences of, and insights into, self-affirmation among individuals living with dementia, their caregivers, and visiting therapy dog handlers. There are also evolving ways to capture the contributions of the therapy dogs themselves to this understanding (e.g., the meaning of canine behavior). Such studies would also allow for critically challenging what is uncovered.

Questions of this sort help remind us that experiencing dementia is not ‘wrong’, and what is important is how we as individuals, caregivers, and communities relate to the illness. This raises the question of whether self-affirmation theory goes far enough to address questions related to how we see each other in our Canadian and ethno culture, and how we relate to difference. This question is not one that self-affirmation theory typically asks. Perhaps the theory could be broadened, to include not only how we adapt when our notion of self is threatened, but also how culture influences the threatening of self-perception. In other words, could self-affirmation theory also include how individuals and communities understand and relate to people experiencing dementia, thereby shifting the focus towards intentional self-affirmation by caregivers and communities? Most importantly, this work can remind us of the power of “the dog doing exactly what the dog does” (Dell), and the possibility for unintentional self-affirmation to become a bit more intentional.

We also suggest more research be carried out on the experiences of the therapy dogs themselves. For example, a recent study suggests therapy dog visiting can be enjoyable and possibly beneficial for the participating dogs [54], which has only been presumed in the past. Levels of salivary cortisol (stress hormone) are maintained or decrease by the end of the visit, indicating no adverse effects for the therapy dog’s wellness [55]. If the therapy dog is not participating fully, and benefitting as well, then the outcomes will be compromised for both.

Some practical suggestions for practice flowing from this case report include adding this material to the “Becoming an Informed Therapy Dog Handler” course in Canada, offered by the University of Saskatchewan [56]. This understanding could also be shared with the family caregivers of individuals living with dementia, as family pets may be involved in the circle of care, and their role could possibly be enhanced. Videos and other forms of animal memories could also be incorporated into the care of people living with dementia, and serve as an impetus to reminisce; one example is the *I Had a Dog* video and worksheet that emerged from Dell and Chalmers’ research [57]. While attention to therapy dogs and animal-assisted interventions generally, and canine-assisted support programs specifically, is becoming better recognized and adopted more by social workers, our work supports Chalmers et al.’s [58] (p. 3) argument for more “specialized education and training on the beneficial impact that companion animals can have on social work practice”. The same applies to other related professions involved in dementia care (e.g., nursing).

And last, perhaps it is also important to flip the dominant dementia narrative in Canada and elsewhere. While one’s sense of dementia-related autonomy or agency may be taken away through restrictions in physical and psychological spaces, one’s autonomy as a human being may be given back, for a moment at least, through interactions with a therapy dog and perhaps, also, the dog handler. If this interaction enables a person to live more autonomously and in the moment, could this also remind all of us, as a community, of the importance of not defining ‘self’ in a limited way. The problem may lie in our expectation of having memories of self in order to have autonomy and worth. People living with dementia are not the problem; rather, the problem may be in how society and caregivers construct dementia, which again brings attention to the importance of what the dog does. The dog is relied upon here and is interpreted as recognizing the person’s profound worth in the moment, and this can contribute to their self-affirmation and well-being.

## 6. Limitations

While our approach to this case report has several merits, limitations are also acknowledged. First, the role of the therapy dog handler in interactions with individuals living with dementia is subtly highlighted, but not fully accounted for. The handlers’ interview responses are also not interrogated, challenged, or critiqued, only reported on, which is the aim of a case report. Further, the therapy dog and handler work together as a cohesive team, and it is difficult to separate the impact of each. We purposefully focused herein on the therapy dog. Second, this article is based on the conversational approach, allowing for reflection of observations and experiences, noting that we must account for our own subjectivities as therapy dog handlers, researchers, and authors. To assist with this, personal biographies are included. Third, the paper is dependent on the memories of the therapy dog handlers. And last, this work is not generalizable.

## 7. Conclusions

This case report reflects on possible connections between the presence of therapy dogs and the presence of moments for self-affirmation in those living with dementia. Our insights align with previous work that shares how therapy dogs are a positive support for people with dementia, and the importance of self-affirmation among people living with dementia. We add unique understanding to this, specifically that self-affirmation may be experienced and enhanced among individuals with dementia through interaction with a canine-assisted support program. We suggest that a more intentional recognition of this is warranted and should be followed up in research and practice attention.

## Data Availability

The focused conversation drawn upon for this case report is available on request from the corresponding author, due to privacy reasons.

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
