# Peer review of "Witnessing Self-Affirming Moments in Persons with Dementia While Interacting with Therapy Dogs: A Case Report"

_animals, 2024, doi:10.3390/ani14243620_

Round 1
Reviewer 1 Report
Comments and Suggestions for Authors
This case report is based on visits by three therapy dogs and their two handlers/researchers to four patients with dementia. The information comes from an approximately 2-hour long focused conversation between the first author and the second and third authors, who also served as the handlers/researchers. My specific comments are detailed below.
Keywords: The terms Therapy dog, dementia, and self-affirmation occur in the title, so will be picked up in searches. Consider replacing these terms with others, such as physical movement, psychological shift, and time transition.
Introduction:
I suggest moving the definition of self-affirmation (experiencing a positive sense of self) from line 66 to line 60, where the term self-affirmation is first used.
Minor point: Line 74, insert “the” before “focus”
Background and literature review:
Lines 149-150: Define “phenomenological hermeneutic method” for readers who may not be familiar with the term (it is not clear from the current description of the research by Swall et al.).
Materials and methods:
The first paragraph (line 171-188) seems out of place and should possibly be pulled out as a separate section, called “Reflexivity statement.” Although personal details for the two handlers/researchers are provided, the same is not done for the first author, who conducted the focused conversation and data analyses and interpretation. The last author, who was involved in validation and editing also is not described. Finally, there is no coverage of potential biases brought by each author to the study and steps taken to mitigate any biases.
Lines 208-211: How much time had elapsed between visits to the four people with dementia and the focused conversation? It would be helpful to provide general time periods for the visits and date for the conversation. Are notes taken by handlers after each therapy dog visit?
Line 211: Please describe the method used by Spee (2005), so readers can understand how the focused conversation occurred without having to look up another paper. Also, I assume the conversation was recorded, but please state this and how it was recorded.
Lines 236-239: Similarly, rather than simply citing papers regarding methods of data analysis and interpretation (references 43 and 45), please also provide a brief description of them because readers should not have to go to other papers to understand how your data were analyzed and interpreted.
Minor point: The word “thematic” is repeated in lines 237-238
Suggested next steps:
Line 584: What is meant by “but also how culture influences the threatening of self-perception.”?
Limitations:
Lines 624-627: Including personal biographies is listed as a step toward acknowledging authors’ subjectivities. However, as mentioned above, personal biographies are not provided for the first and last author, and potential biases are not described.
Reviewer 2 Report
Comments and Suggestions for Authors
This paper provides a qualitative analysis of two dog handler/researchers’ experiences with four people with dementia, in a visitation dog program. It is a useful contribution to the literature. I have a few suggestions before I can recommend for publication.
Ethics waiver – could the authors please provide detail about the original research projects that received ethics approval, which then meant that the approval for this one could be waived?
L220 – please cite the claim about witness vs observe, as well as to explain what the difference is. I’m also not certain it is appropriate to place data (i.e., a quote) into the methods section.
Throughout – the language around the program itself varies a lot. Sometimes the authors refer to canine-assisted interventions, sometimes animal-assisted interventions, sometimes therapy dog interventions, and sometimes animal-assisted support programs. The first two are not recommended for use according to the Binder et al paper that the authors cited when defining animal-assisted support programs. I think the latter is the most appropriate, so I recommend using that one and sticking with it when discussing their own program; canine-assisted support programs would also be fine. In the section on canine-assisted interventions (L121), and in the Discussion, I recommend using animal/canine-assisted treatment or support program, as relevant.
Limitations – I suggest adding something about the handlers having an interest in ensuring that the program is successful, and this is a limitation. More ‘objective’ observations would be warranted in the future. Similarly, I am not certain that the authors can declare no conflict of interest since two of them are researcher, participant, and dog handler. There may be no FINANCIAL interest, but I don’t see how they can argue there is no interest of any kind. This should be acknowledged, and an explanation given of how the roles were maintained separate, to the extent possible.
Reviewer 3 Report
Comments and Suggestions for Authors
This is a very well written manuscript. I enjoyed reading it. The manuscript is exceptionally well grounded in literature, and sets out a strong framework and a novel area of investigation and contribution. However, I do have some comments and concerns both ethical and epistemological about the possible scope of this piece in its ability to successfully make claims about the subjective experiences of people with dementia.
Firstly, I am somewhat uncomfortable with the idea that “ethics approval was not required”. I also feel this is a decision for an IRB, rather than the authors themselves (if this was the case, this should be specified). The authors are reporting their observations and interactions with vulnerable humans. Did the people affected by dementia know they were to be the subject of research?
Similarly, the article feels a little confused in what it is reporting. It is sold to readers as being a “focused conversation” between two researchers, hence not needing ethics, yet the very first line of the first results theme states that “visits were often in spaces viewed by the persons with dementia as being less restrictive and controlled”. This is either reporting the perspectives of humans which would have required research with human subjects, or is attributing views to, and speaking for, a vulnerable group – it feels a little dehumanizing to speak of and for people in this way.
Likewise, with the focus of this manuscript on “self-affirmation” – I’m not sure of the potential for any conclusions to be drawn without involving those “selves” directly. What is reported is the researchers’ perceptions of people’s experiences, rather than subjective experiences – I think this is a huge challenge at the heart of this manuscript requiring more critical reflection about what can be known and claimed from the ‘data’ invoked. I feel the paper goes too far and oversteps in the claims and framing of what is reported and made knowable at many points throughout the manuscript. Reporting on facilitators/therapists perspectives is fine and valid, but I struggle with the idea that those delivering the intervention have insight into the inner-worlds and subjective experiences of those participating in the intervention – such would require qualitative research with those people to determine if they do feel “self-affirmation”, rather than speaking for them and attributing experiences. Otherwise, this paper needs to take a step back, and be more honest about framing itself as exploring and reporting the views of handlers as to what they believe they are delivering and why they believe it to be beneficial, rather than attributing views that the participants in the intervention found it “was self-affirming with respect to how individuals saw themselves”.
I recognize that the authors and handlers involved in this piece have a wealth of experience and are experts at what they do and have seen up-front the benefits of therapy dog interventions – and there is more background and history to this piece than what is described in the methodology. However, this needs more careful and nuanced framing in how it is reported in this piece, as it too often crosses the line into making claims about the experiences of people with dementia without explicitly asking them themselves. People with dementia as a group are too often ‘spoken-for’, so this is a particularly important nuance to get right here.
Broadly, the manuscript is also very positive and affirmative. There are opportunities for further reflection about who is excluded through and by the use of therapy dogs, the challenges of utilizing therapy dogs with people affected by dementia, and the moments were interventions are unsuccessful or even negative.
Similarly, given the role of the authors in delivering the intervention and being advocates for therapy dog programs, I do think the manuscript may need to reflect a little on potential for bias.
Reviewer 4 Report
Comments and Suggestions for Authors
Line 52- reword this sentence. Something more like: “for those impacted, there are also hopeful..” and maybe separate or break up the sentence because it runs on in lines 53 and 54.
Line 74- I think there is a word missing after “being”. Maybe “the”
Paragraphs 72-85: is this similar to animal assisted activities? I know there aren’t universal agreements on terms for Animal Assisted Interventions but it may be helpful to use resources such as IAHAIO https://iahaio.org/best-practice/white-paper-on-animal-assisted-interventions/ and AAII https://aai-int.org/aai/standards-of-practice/ for most commonly used terms
Lines 144-145: looks like something is missing after “e.g.”
Methods & Materials section: Review format for numbers in sections (i) vs 1.
Line 529: Is there supposed to be a p inf ront of 101?
Line 548: Reword “define what self in dementia may mean”. Sounds a little confusing.
Line 555: Review in text citation here for direct quote
Lines 577 & 606: big spaces in between words on pdf version
Limitations: Though valuable research, could you also say that this information is not generalized since it was a small participant group and qualitative?
I think it is also worth exploring how to maintain welfare of animals and people and ensuring safety when working with populations that are higher risk.
Round 2
Reviewer 1 Report
Comments and Suggestions for Authors
The authors have revised their manuscript in line with my earlier comments (e.g., defining terms, providing more detail in the Methods section regarding the reflexivity statement, timing of visits and interview, and approaches taken, such as ORID). I also appreciate the additions to the Limitations section regarding reliance on memories of therapy dog handlers and that the work is not generalizable. In short, the manuscript is much improved.
Author Response
Thank you again for your thoughtful review and very useful comments. We appreciate your time and your commitment to reviewing this work.
Reviewer 3 Report
Comments and Suggestions for Authors
I appreciate the authors detailed response to my previous questions, and the work they have done to thoughtfully engage with some quite challenging questions. I am grateful for the sensitivity they have introduced here, and their constructive approach to re-working their manuscript.
I thought that the authors responded very thoughtfully about their decisions around ethics in the response to the reviewers. It would be useful to see some of that critical reflection incorporated into the manuscript itself, to assure other readers of the background and context behind these decisions.
My main challenge, having read this revised version, is that I am now somewhat confused as to what really makes this a ‘case report’ – the language of which the authors use frequently in the manuscript. I think it is an increasing stretch to claim that “the paper presents the cases of four individuals diagnosed with dementia” (p2, line 65) – particularly after the revisions. This is in direct opposition to what the authors assure us of in their response that “we believe we have been very transparent from the abstract through to the conclusion that the paper focuses on the therapy dog handlers’ insights” – we are promised cases. I appreciate the work the authors have started here, but I think it needs to go further in thinking about what is promised and how things are framed. And/or some critical interrogation of what is meant by the format of the ‘case report’ to show how and why this is within the format and remit of one.
Additionally, there is a tension between the language the authors have introduced to resolve my previous point about potential epistemological claims, and the format of the case report: possibly, could, seemed, potentially, appeared, seemingly… etc. I think such is useful (to a point) and understand why this hesitancy and distance has been introduced, and feel guilty in in saying this (as this change is in response to my previous query), but it undermines the manuscript and causes a reader to question just what is being reported here, if all is hypothesized and conjectural? Little focus is given to ‘the cases’ themselves. At present, the analytical unit of the report is the two interviewed authors, not the cases. I think more needs to be done to stress the links to this format.
As an aside, I wondered if the authors had seen: Sori and Hughes. "Animal-assisted play therapy: An interview with Rise VanFleet." The Family Journal 22.3 (2014): 350-356. – and whether taking some inspiration from such a format might be an interesting way of framing some of this, rather than the conventions of the ‘case report’.
The language choice referenced above also connects to my point about a discussion of bias. I don’t think adding some author bios quite tackles this comment, as I was after a discussion of how bias was accounted for. Indeed, is it not altogether unsurprising that people with such backgrounds would report that therapy dogs “can potentially enhance self-affirmation” or that such “could possibly allow for a decrease in some common dementia symptoms”? Again, there is something of a core methodological and epistemological challenge here. This needs to be accounted for much more directly. The interviewee responses are not interrogated, challenged, or critiqued, only reported.
I think the manuscript is much improved, but still requires some further reflection to ensure it is balanced and consistent in messaging.
